# A Pilot Study of Circulating Monocyte Subsets in Patients Treated with Stem Cell Transplantation for High-Risk Hematological Malignancies

**DOI:** 10.3390/medicina56010036

**Published:** 2020-01-18

**Authors:** Ida Marie Rundgren, Elisabeth Ersvær, Aymen Bushra Ahmed, Anita Ryningen, Øystein Bruserud

**Affiliations:** 1Department of Biomedical Laboratory Scientist Education and Chemical Engineering, Faculty of Engineering and Natural Sciences, Western Norway University of Applied Sciences, 5020 Bergen, Norway; imru@hvl.no (I.M.R.); elisabeth.ersver@hvl.no (E.E.); anita.ryningen@hvl.no (A.R.); 2Department of Clinical Science, University of Bergen, 5020 Bergen, Norway; 3Section for Hematology, Department of Medicine, Haukeland University Hospital, 5021 Bergen, Norway; aymen.bushra.ahmed@helse-bergen.no

**Keywords:** monocytes, leukemia, stem cell transplantation, flow cytometry, hematology

## Abstract

*Background and Objectives:* Autologous and allogeneic stem cell transplantation is used in the treatment of high-risk hematological malignancies, and monocytes are probably involved in hematological reconstitution as well as posttransplant immunoregulation. The aim of our study was to investigate the levels of circulating monocyte subsets in allotransplant recipients. *Materials and Methods:* The levels of the classical, intermediate, and nonclassical monocyte subsets were determined by flow cytometry. Sixteen patients and 18 healthy controls were included, and the levels were analyzed during pretransplant remission (*n* = 13), early posttransplant during cytopenia (*n* = 9), and early reconstitution (*n* = 9). *Results:* Most patients in remission showed a majority of classical monocytes. The patients showed severe early posttransplant monocytopenia, but the total peripheral blood monocyte counts normalized very early on, and before neutrophil and platelet counts. During the first 7–10 days posttransplant (i.e., during cytopenia) a majority of the circulating monocytes showed a nonclassical phenotype, but later (i.e., 12–28 days posttransplant) the majority showed a classical phenotype. However, the variation range of classical monocytes was wider for patients in remission and during regeneration than for healthy controls. *Conclusions:* The total peripheral blood monocyte levels normalize at the very early stages and before neutrophil reconstitution after stem cell transplantation, and a dominance of classical monocytes is reached within 2–4 weeks posttransplant.

## 1. Introduction

Acute myeloid (AML) and acute lymphoblastic leukemia are both aggressive malignancies characterized by the accumulation of immature malignant cells in the bone marrow [1,2]. However, several less aggressive hematological malignancies are also regarded to have unfavorable prognoses with short expected survival, e.g., the high-risk myelodysplastic syndromes [3] and certain prolymphocytic leukemia variants [4]. The only or the best possibility for cure for all these malignancies is intensive conventional chemotherapy, possibly combined with allogeneic or autologous stem cell transplantation [1]. However, several new and promising therapeutic approaches are now considered and/or are available for high-risk hematological malignancies, including T-cell targeting immunotherapy [5,6], new monoclonal antibodies for the treatment of acute lymphoblastic leukemia [7], modulation of apoptotic regulation with increased proapoptotic activity, or [8] inhibition of intracellular signaling, including metabolic targeting [9,10]. These new therapeutic strategies may be used as a part of the initial treatment to reduce the risk of later relapse and thereby reduce the need for high-toxicity antileukemic therapy (e.g., allogeneic stem cell transplantation), or they can be used to reduce the risk of posttransplant relapse [9,11].

The median age for the time of diagnosis is 60–70 years for most hematological malignancies [1], and the most intensive therapeutic strategies can only be used for relatively young patients without severe comorbidity [12]. Conventional intensive chemotherapy is usually considered for patients up to 75–80 years of age [13], whereas stem cell transplantation is usually considered for patients up to 70–75 years of age [12,14]. However, the use of these most intensive therapeutic strategies in elderly patients will not only depend on the age, clinical evaluation, and comorbidity score of the individual patient, but will also differ between institutions. For most patients, stem cell transplantation is therefore used as a consolidation therapy after they have reached complete hematological remission (i.e., disease control without morphological signs of leukemia) in response to the initial induction chemotherapy [1,14]. The strong antileukemic effect of stem cell transplantation is caused by the intensive conditioning therapy and, especially for allotransplant recipients, antileukemic immune reactivity mediated by graft immunocompetent cells [15].

Monocytes constitute up to 10% of peripheral blood leukocytes in healthy individuals [16]. They can be differentiated into macrophages and dendritic cells [17], in the endothelial direction [18] and possibly also in the direction of monocytic myeloid-derived suppressor cells [19,20]. Based on their expression of the two cell surface receptors CD14 and CD16, monocytes are divided into classical (CD14^bright^CD16^negative^), intermediate (CD14^bright^CD16^dim^), and nonclassical (CD14^dim^CD16^bright^) subsets [21,22,23]. CD16 is a low-affinity IgG receptor that can initiate intracellular signaling, and it is thereby important for regulation of monocyte cytotoxicity [24]. CD14 is a pattern recognition receptor; it functions as a co-receptor for Toll-like receptors but has also several functions independent of these receptors, including transport of inflammatory lipids to induce phagocytosis [25]. Thus, the functional heterogeneity of various monocyte subsets is reflected by the two molecular markers used for identification of the three monocyte subsets, and both markers are involved in the regulation of important phenotypic characteristics through their modulation of intracellular signaling. Classical monocytes often constitute at least 90% of circulating monocytes in healthy individuals [21,23,26].

The monocyte subset levels during the first 4 weeks posttransplant have not previously been characterized in detail, but a few studies have investigated later monocyte reconstitution, from day +28 until day +100 posttransplant. Firstly, both monocytes and neutrophils show early posttransplant reconstitution [27,28,29], and total monocyte reconstitution occurs early, both after myeloablative and reduced intensity conditioning [28]. Total monocyte reconstitution also seems to occur early in patients receiving haploidentical transplantation for nonmalignant bone marrow disorders [30]. Secondly, the levels of circulating CD14^+^CD16^+^ monocytes (i.e., intermediate and nonclassical monocytes) after day +30 posttransplant are associated with decreased incidence of chronic graft versus host disease (GVHD) after allotransplantation [20]. Thirdly, another study described an association between relatively high levels of classical monocyte (i.e., CD14^+^CD16^−^) levels early posttransplant, as well as improved survival, relapse risk, and transplant-related mortality. These associations seem to be maintained during the first 100 days posttransplant [31]. The patients included in this last study were heterogeneous and half of them received umbilical cord blood grafts [31].

The peripheral blood levels of several immunocompetent cells are altered after stem cell transplantation, and the posttransplant CD4^+^ T-cell defect can last for months [27]. Our hypothesis was that the levels of the various circulating monocyte subsets are also altered, especially during the early (i.e., first 4 weeks) posttransplant period, even though the total monocyte levels normalize at the early (i.e., before the neutrophil and platelet counts) stages of this period. The immunoregulatory events during the first weeks posttransplant are important for outcome after stem cell transplantation (for detailed discussion and additional references see [15]), and our aim was therefore to characterize circulating monocyte subset levels during this period. We used a highly standardized methodology to characterize peripheral blood levels of monocyte subsets for a relatively homogeneous and unselected group of patients (i.e., all having hematological malignancies with adverse prognosis) receiving peripheral blood stem cell transplantation. All patients were transplanted after reaching complete hematological remission, and our studies of circulating monocyte subsets included pretransplant levels (i.e., after reaching remission), early levels during severe posttransplant pancytopenia, and levels during hematological reconstitution.

## 2. Materials and Methods

### 2.1. Patients and Healthy Controls

The study was conducted according to the Declaration of Helsinki. All samples were collected after written informed consent (Regional Ethics Committee REK Vest 2015/1759), and the use of biological material in the present project was also approved (REF Vest 2017/305, 2013/102). Control samples were derived from 18 healthy blood donors (7 females and 11 males, median age 53 years with range 21–73 years). In accordance with the approved routines at the Blood Bank, Haukeland University Hospital, peripheral venous blood samples for medical research were donated after written informed consent. Our hospital is the only center providing intensive antileukemic treatment in a defined geographical area of Norway, and our patients represent a consecutive group of patients receiving intensive antileukemic treatment/stem cell transplantation during an eight month period. Our study should therefore be regarded as population-based.

All our patients received initial chemotherapy to achieve disease control (Table 1, Appendix A). Thus, we investigated peripheral blood levels of monocyte subsets for stem cell recipients who had achieved complete hematological remission, i.e., normal levels of immature/abnormal cells in the bone marrow judged by light microscopy, peripheral blood neutrophils > 1.0 × 10^9^/L and peripheral blood platelet counts > 100 × 10^9^/L [1]. One myelodysplastic syndrome (MDS) patient reached complete remission with incomplete peripheral blood normalization. In addition, patients with lymphoproliferative disease did not show flow cytometric evidence of minimal residual bone marrow disease, and imaging studies did not show evidence of residual disease immediately before stem cell transplantation.

Sixteen patients were included in the various parts of our study (see Table 1, right part), but only 11 of these patients were treated with stem cell transplantation. They were all treated at Haukeland University Hospital during the period June 2016–June 2017. Patients were transplanted after reaching complete hematological remission, and they received G-CSF mobilized peripheral blood stem cell grafts. All our allotransplant recipients received grafts from HLA (human leukocyte antigen)-identical sibling donors, the same GVHD prophylaxis (cyclosporine plus methotrexate) [32], VOD prophylaxis (i.e., ursodeoxycholic acid) [32] and pretransplant trimethoprim–sulfamethoxazole treatment, but no fungal prophylaxis. Five patients were not allotransplanted for the following reasons: because they had a favorable karyotype (one patient), were unfit for stem cell transplantation (two patients), had a second relapse before transplantation could be performed (one patient), or had been transplanted at another hospital (one patient).

### 2.2. Blood Sampling, Flow-Cytometric Analysis of Monocyte Subsets and Analysis of Peripheral Blood Leukocytes

Peripheral venous blood was drawn in ACD-A blood sampling tubes (#248368, BD Biosciences, San Jose, CA, USA). All samples were collected between 08:00 and 10:00 a.m. and were processed at room temperature within 120 min. Sampling was conducted at the same standardized time interval because total leukocytes show diurnal variations [33,34,35,36]. Four milliliters of the anticoagulated blood and 46 mL of lysis buffer (#55589, BD Biosciences) were mixed and incubated for 15 min at room temperature. Subsequently, leukocytes were collected by centrifugation (400× *g*, 5 min, room temperature) and thereafter washed in phosphate-buffered saline with 1% bovine serum albumin (BSA, Bovine serum albumin fraction V, #10735086001, Sigma-Aldrich/Merc KGaA, Darmstadt, Germany). The cells were reconstituted in 200 µL 1% BSA/PBS with 10% immunoglobulin solution (Octagam 100 mg/mL, Octapharma, Lachen, Switzerland). The following mouse anti-human antibodies were used (all from BD Biosciences): CD14 Alexa 488 (Clone M5E2), CD56 Alexa 647 (Clone B159), CD16 PerCpCy™5-5 (Clone 3G8), CD45 V500 (Clone HI30), CD11b V540 (Clone ICRF44 (44)) and HLA-DR PE (Clone G46-6). The staining procedure and gating strategy for identification of monocytes and monocyte subsets has been previously described in detail [29,37], and all samples were analyzed using a 10-parameter BD FACS Verse flow cytometer equipped with 404, 488, and 640 nm lasers.

Comparison of peripheral blood levels of neutrophils and total monocytes was based on analyses using an accredited clinical hemocytometer (Laboratory for Clinical Biochemistry, Section for Hematology, Haukeland University Hospital).

### 2.3. Statistical Analyses

We applied IBM SSP statistics 23 for all analyses. The Kruskal–Wallis test was used for comparison when several groups were included in the analysis. The Wilcoxon’s rank sum test was used for comparison of differences between two different groups of individuals, whereas the Wilcoxon’s test for paired samples was used for statistical comparison of two different observations in the same patients. *p*-values below 0.05 were regarded as statistically significant.

## 3. Results

### 3.1. The Pretransplant Status of Our Patients with Hematological Malignancies; Monocyte Subsets for Patients in Complete Hematological Remission after Conventional Chemotherapy

Patients are usually treated with initial conventional chemotherapy to achieve disease control (see Section 2.1) [1]. We analyzed the monocyte subset levels after 13 different chemotherapy cycles in 11 patients (Table 1) who fulfilled the criteria of disease control/complete remission at the time of sampling, and in addition they all had peripheral blood total monocyte counts within the normal range. The results are presented in Figure 1. The monocyte subset levels showed a wider variation in the patients than in the healthy controls, but for 11 of the 13 samples, the majority of circulating monocytes (>80%) belonged to the classical monocyte subset. For the last two patients, classical monocytes constituted <20% of total monocytes. When comparing the overall results, the percentages of classical, intermediate, and nonclassical monocytes for the remission patients did not differ significantly from the healthy controls. The two exceptional patients were patients 1 and 16 (Table 1). For five of the patients, we also estimated the peripheral blood concentrations of the various monocyte subsets, and both classical (*p* = 0.002), intermediate (*p* = 0.002), and nonclassical monocyte (*p* = 0.006) concentrations were significantly decreased compared with the corresponding concentrations in the healthy controls.

### 3.2. The Peripheral Blood Levels of Various Monocyte Subsets Show Wide Variation during the Period of Severe Posttransplant Pancytopenia

The levels of circulating monocyte subsets during the period of severe posttransplant pancytopenia were investigated for nine transplant recipients (Figure 1). Severe pancytopenia was then defined as total peripheral blood leukocytes ≤ 0.3 × 10^9^/L, neutrophils < 0.2 × 10^9^/L, total monocytes < 0.1 × 10^9^/L, and dependency of regular platelet transfusions to keep the platelet count above 10–20 × 10^9^/L. Two of these patients had overly low peripheral blood leukocyte counts to allow for reliable estimation of monocyte subset distribution (Table 1, patients 2 and 11). The two exceptional patients with detectable but low (i.e., <3%) levels of classical monocytes were the two allotransplant recipients, patients 1 and 8. The other seven patients showed a distribution within the variation ranges of the normal controls. Thus, the transplant recipients show a wider variation than the healthy controls in venous blood monocyte subset levels during the period of severe posttransplant pancytopenia. The percentage of intermediate monocytes was even significantly decreased compared with healthy controls (only the seven patients with detectable levels included, *p* = 0.002), even though healthy controls also show levels for this subset. By contrast, the percentages of classical and nonclassical monocytes did not differ significantly from the healthy controls.

For two of the patients, we estimated the venous blood concentrations of the three monocyte subsets and, as expected, both classical, intermediate, and nonclassical monocytes for these two patients showed peripheral blood concentrations outside the corresponding variation ranges of the healthy controls.

We investigated the peripheral blood levels of the three monocyte subsets for patients with severe pancytopenia, as defined above, after conventional intensive antileukemic chemotherapy (seven patients examined). These patients also showed a wide variation in their peripheral blood levels of both classical (range < 0.01–100%), intermediate (range < 0.01–42%), and nonclassical monocytes (range < 0.01–22%). Thus, a wide variation in peripheral blood levels during treatment-induced cytopenia is observed not only for stem cell transplant recipients, but also for patients receiving conventional intensive antileukemic chemotherapy.

### 3.3. Stem Cell Transplant Recipients Show Early Posttransplant Total Monocyte Reconstitution

Previous studies have demonstrated that stem cell recipients show early reconstitution of total monocytes, and this is true for both allotransplant [18,19] and autotransplant recipients [20]. All our stem cell transplant recipients showed early reconstitution with normalized levels of circulating monocytes before the normalization of the neutrophil counts (Figure 2, Appendix A). Furthermore, many of the recipients showed a posttransplant period of increased levels of circulating monocytes; this could be observed for eight of our 12 stem cell transplanted patients, including six out of the nine allotransplant recipients. The maximal total monocyte levels for our patients were reached between day +21 and +35 posttransplant, increased levels were seen both for patients receiving autologous and allogeneic stem cell transplantation, and the maximal levels varied between 1.41 × 10^9^/L and 5.85 × 10^9^/L (normal range 0.04–1.30 × 10^9^/L).

### 3.4. Early Posttransplant Monocyte Subset Regeneration after Stem Cell Transplantation

We investigated the posttransplant peripheral blood levels of the three monocyte subsets during early regeneration when the total monocyte counts had reached normal levels (normal range 0.04–1.3 × 10^9^/L). Nine patients were tested 12–28 days posttransplant when monocyte counts had normalized (median concentration 0.74 × 10^9^/L, range 0.45–1.66 × 10^9^/L). At the time of testing, five of the patients were still neutropenic; the patients had median neutrophil levels of 1.6 × 10^9^/L with a variation range of 0.5–2.8 × 10^9^/L (lower normal limit 1.7 × 10^9^/L). The levels of the three monocyte subsets are presented in Figure 1. The majority of circulating monocytes early after posttransplant reconstitution belonged to the classical monocyte subset for all patients (median 76%, range 58–100%), but the variation range was wider for the regenerating patients than for the healthy controls. The levels of both intermediate and nonclassical monocytes were below 20%. The percentages of classical (decreased, *p* = 0.001), intermediate (increased, *p* = 0.001), and nonclassical monocytes (increased, *p* = 0.003) for patients showing hematological regeneration differed significantly from the healthy controls. However, the variation between patients was smaller during regeneration than for patients in pretransplant remission.

We investigated the levels of circulating monocyte subsets at various time points both during and soon after the period of severe posttransplant cytopenia for nine stem cell transplant recipients. The results for three patients are presented in Figure 3. It can be seen that the percentage of classical monocytes was initially decreased, and could be very low before it increased after 2–4 weeks. It was also observed that the intermediate monocytes constituted a minority during the whole period but showed a wider variation range than the healthy controls, whereas the nonclassical monocytes showed high/increased levels during the first week, but after 4 weeks, they were a small minority. All nine patients showed a similar pattern for the classical monocytes with initially low/decreased levels with a later increase, until after 4 weeks, they constituted a majority of the circulating monocytes.

For four of the patients, we estimated the peripheral blood concentrations of the three monocyte subsets, and classical (*p* = 0.002), intermediate (*p* = 0.008), and nonclassical monocyte (*p* = 0.006) concentrations differed significantly from the levels in the 18 healthy controls.

## 4. Discussion

Several observations suggest that monocytes are important for outcome after allogeneic stem cell transplantation. Firstly, differences in the amounts of graft monocytes between younger and older stem cell donors may contribute to the adverse prognosis when using older stem cell donors [38,39]. Secondly, monocytes seem important for the development of posttransplant tolerance [40]. Finally, the monocyte-lymphocyte ratio seems to have a prognostic impact at least in haploidentical transplantation [41]. In our present study, we therefore investigated the balance between classical, intermediate, and nonclassical monocytes after stem cell transplantation in patients with high-risk leukemia or MDS. To the best of our knowledge, this is the first study of early monocyte subset reconstitution after stem cell transplantation for hematological malignancies. We observed very early reconstitution of total monocytes, and also the dominating classical monocyte subset after intensive conditioning therapy followed by stem cell transplantation.

In our present study, we focused on the early posttransplant period, i.e., the first 4 weeks after stem cell reinfusion. This period is important for outcome after transplantation; this is illustrated by the observations that both the type of conditioning therapy as well as the use of hematopoietic growth factors during this period influence the risk of severe posttransplant complications [15]. We investigated a group of consecutive patients with high-risk hematological malignancies admitted to our institution for intensive chemotherapy. All patients had hematological malignancies with adverse prognoses, and received intensive chemotherapy, including stem cell transplantation for 11 of them. We included all admitted patients without any selection, and for this reason, we investigated patients with different diagnoses who had received different forms of pretransplant conditioning therapy. However, all patients received peripheral blood mobilized stem cell grafts; most of them received allografts (all from HLA-matched siblings) but a minority of our patients received autologous stem cell grafts. The pattern of reconstitution seemed to be similar for allotransplant and autotransplant recipients, which is not surprising as autotransplanted patients received the same conditioning chemotherapy as allotransplanted patients.

Our present study included a small number of patients, but our patients are relatively homogeneous. We would also emphasize that our patient cohort represents a consecutive group of patients from a defined geographical area during a defined time period. Thus, our patients should also be regarded as unselected.

All transplant recipients included in our study had severe hematological malignancies, and they initially received intensive conventional chemotherapy with the intention to induce complete hematological remission. They all reached stable complete remission prior to transplantation. Patients receiving such intensive conventional chemotherapy showed a similar wide variation in monocyte subset levels during treatment-induced cytopenia to the patients with posttransplant cytopenia, even though they had received repeated cycles of intensive consolidation treatment with relatively short intervals after remission induction [1].

Previous studies have demonstrated that monocyte reconstitution (i.e., normalization of peripheral blood monocyte levels) occurs relatively early after allogeneic stem cell transplantation [37], and this was also true for all our allotransplant and autotransplant recipients. Normalization of the circulating monocyte levels usually occurred before normalization of the neutrophil levels. Thus, even though our study included relatively few patients, they should be regarded as representative because they show the expected early normalization of both total monocyte and neutrophil peripheral blood levels.

We have previously investigated the levels of circulating monocyte subsets in myeloma patients receiving autologous stem cell transplantation [29]. These myeloma patients differ from the patients included in our present study as (i) they received different and less intensive pretransplant treatment without remission induction; (ii) the intention of the transplantation is stabilization but not cure; and (iii) they received less intensive conditioning therapy as well as growth factor treatment posttransplant. The myeloma patients also showed early monocyte reconstitution, but displayed wider variations in the monocyte subset levels both pretransplant and immediately after transplantation during cytopenia. However, despite these differences, both patient groups showed early monocyte reconstitution, and a majority of circulating monocytes show classical phenotypes within 4 weeks, both for the majority of myeloma patients and all the present leukemia patients.

The CD14 and CD16 markers used for identification of monocyte subsets reflect the functional heterogeneity of the three subsets because both these surface molecules initiate downstream intracellular signaling that is involved in the regulation of important functional characteristics, including proinflammatory and phagocytic activity [24,25]. Thus, our present results show that the balance between functionally different monocyte subsets can be altered not only immediately before, but also during the early posttransplant period (i.e., the first 4 weeks) when compared with later after transplantation.

Several previous observations suggest that immunological events during the early posttransplant period are important for outcomes after stem cell transplantation, especially allogeneic transplantation. The effect of G-CSF therapy on outcome after allogeneic stem cell transplantation has been investigated in two large studies. In the European study, G-CSF therapy was associated with decreased overall survival due to increased non-relapse mortality (i.e., severe GVHD) [42], whereas in another study, G-CSF therapy did not influence survival [43]. A major difference between the two studies was the increased use of total body irradiation in the European study. An experimental study has previously shown that total body irradiation is associated with an increased activation of dendritic cells, which can also be differentiated from monocytes, and thereby increased proinflammatory alloreactivity [44]. Furthermore, the intensification of GVHD prophylaxis during the first one or two posttransplant weeks with methotrexate or cyclophosphamide also demonstrates that immunological events early posttransplant are important for outcome [45,46,47], and that this intensification may not only affect the T-cells, but also directly influence the monocytes [48,49]. Taken together, all these observations suggest that the altered balance between various monocyte subsets early after allotransplantation may be important for the later outcome. Other studies suggest that this may also be true for autologous stem cell transplantation [50].

Our present study shows that patients with hematological malignancies have a wider variation in the relative levels (i.e., percentage) of circulating monocyte subsets than healthy individuals during periods of antileukemic therapy. However, this variation is reduced after stem cell transplantation due to rapid posttransplant reconstitution of total and classic monocytes, although there is still a wider variation between patients after transplantation. Previous studies have shown that later posttransplant variations in monocyte subset are associated with outcome after transplantation [20,31]. Other investigators have observed that clinical events during the very early period after stem cell transplantation are important for outcome/survival [15]. Future studies should therefore try to clarify whether the early variations described in our present study are also associated with outcome after transplantation, and if so, whether these early differences between patients may be used as clinically relevant biomarkers and a basis for early interventions to improve outcome. These studies should possibly also investigate the balance between remaining recipient and donor monocytes in allotransplant recipients [44]. Finally, previous studies have demonstrated that the role of monocytes in inflammation may be influenced by other leukocyte subsets, and the interactions between monocytes and other normal leukocytes during the early posttransplant period also need to be investigated [51,52].

## 5. Conclusions

Our present data show that monocytes reconstitute early after stem cell transplantation for high-risk leukemia and MDS patients. Our data also show a greater variation in the distribution of monocyte subpopulations among patients during both pretransplant stable remission and posttransplant pancytopenia, whereas the variation is less 4 weeks posttransplant, although still larger than for healthy controls. Thus, our present studies show an expected normalization of total monocyte counts early after stem cell transplantation, but our results also show that normalized total monocyte counts do not mean a normalization of monocyte functions (i.e., normalization of the balance between functionally different monocyte subsets).

## Figures and Tables

**Figure 1 medicina-56-00036-f001:**
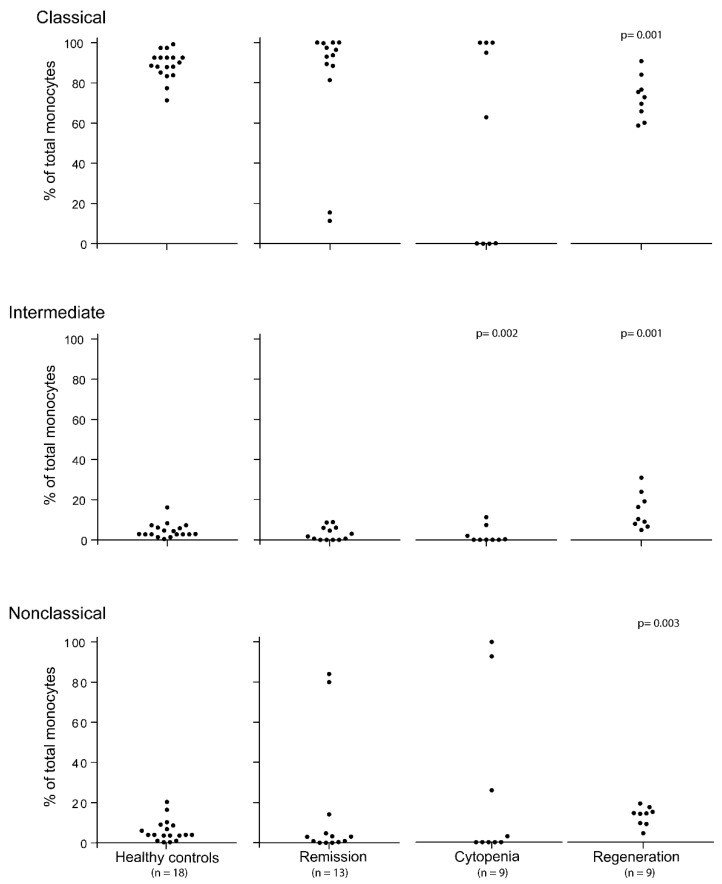
Peripheral blood levels of the classical, intermediate, and nonclassical monocyte subsets in patients receiving intensive chemotherapy for high-risk hematological malignancies. The figure presents the levels for healthy controls (left) and leukemia patients in stable complete hematological remission (i.e., no morphological signs of leukemia, neutrophils > 1.0 × 10^9^/L, and independence of platelet transfusions; all patients, in addition, had normal total monocyte counts) (middle left), patients with severe posttransplant pancytopenia (neutrophils < 0.2 × 10^9^/L, dependency of platelet transfusions, total monocytes < 0.2 × 10^9^/L; middle right), and during posttransplant regeneration (i.e., normal total monocyte counts, increasing neutrophil counts > 0.5 × 10^9^/L; right). Those two patients without detectable monocyte levels during cytopenia are marked among the lowest percentages for all three monocyte subsets. Statistical analysis using the Kruskal–Wallis test showed significant variations for classical (*p* = 0.029), intermediate (*p* = 0.001), and nonclassical monocytes (*n* = 0.03). The *p*-values for statistically significant differences between healthy controls and individual patient/monocyte subset combinations (Wilcoxon’s test for paired samples) are indicated at the top of each of these figures. The results are presented as the percent among total monocytes in peripheral venous blood.

**Figure 2 medicina-56-00036-f002:**
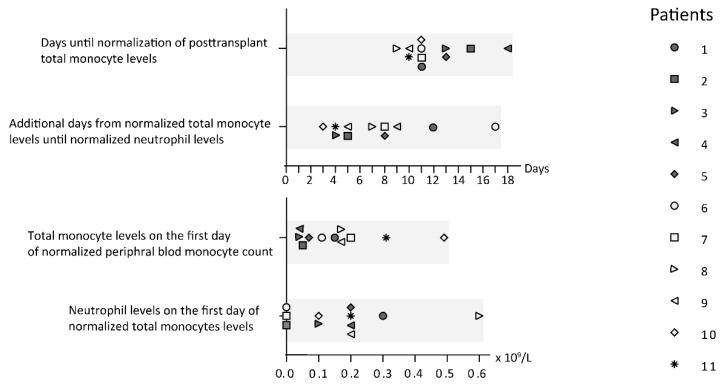
Monocyte and neutrophil reconstitution after conditioning therapy followed by stem cell transplantation. All 11 stem cell transplant recipients (patients 9–11 received an autologous stem cell graft) were included in this part of the study; the symbols for each of the patients are indicated to the right in the figure. The upper part of the figure shows the first posttransplant day with normal total monocyte cell counts (normal level 0.4–1.3 × 10^9^/L; day 0 being the day of stem cell infusion) and the number of additional days until normalized peripheral blood neutrophil counts (lower normal limit of neutrophils 1.7 × 10^9^/L). The lower part of the figure presents the peripheral blood concentrations of total monocytes and neutrophils on the first day with normalized total monocyte counts in the blood. All samples were collected between 07:00 and 08:30 a.m.

**Figure 3 medicina-56-00036-f003:**
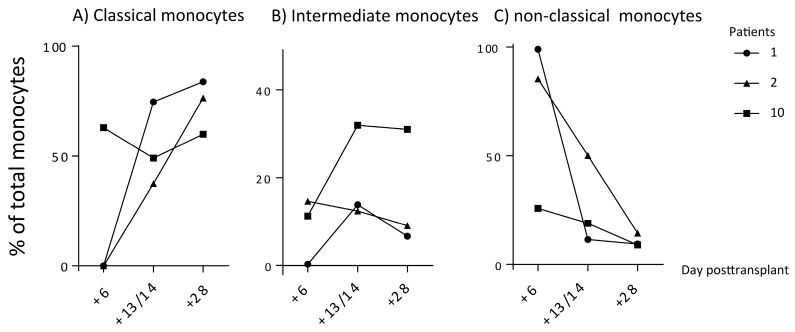
Posttransplant peripheral blood levels of classical (A), intermediate (B), and nonclassical monocytes (C) in three leukemia patients. The figure presents the levels for days +6, +13/+14, and day +28 posttransplant for patients 1, 2, and 10 (see Table 1). These three time points after the stem cell infusion correspond to severe cytopenia and early posttransplant hematological regeneration with normalization of monocyte counts and, finally, also normalized neutrophil counts. The results are presented as the percentage of each monocyte subset among total monocytes. The day of testing is indicated on the *x*-axis; day 0 is the day of stem cell infusion.

**Table 1 medicina-56-00036-t001:** Clinical characteristics of the patients included in the study and the use of blood samples in the various parts of the study. Age is given in years. The time when tested is indicated in the right part of the table (CC, cytopenia after conventional chemotherapy; REM, remission after chemotherapy; CSCT, cytopenia after stem cell transplantation; REC, reconstitution).

Id	AGE	SEX	Diagnosis	Treatment	Treatment	CCC	REM	CSCT	REC
**Conventional Induction and Consolidation Followed by Stem Cell Transplantation**									
1	29	M	ALL	Remission induction followed by allotransplantation	Conditioning with busulfan plus cyclophosphamide		+	+	+
2	61	F	MDS-HR2	Remission induction followed by allotransplantation	Conditioning with fludarabine and treosulfan		+	+	+
3	67	M	PLL-T	Remission induction followed by allotransplantation	Conditioning with fludarabine and treosulfan				+
4	64	M	MDS-HR2	Remission induction followed by allotransplantation	Conditioning with fludarabine and treosulfan			+	+
5	54	F	AML-MDS	Remission induction followed by allotransplantation	Conditioning with busulfan plus cyclophosphamide		+		+
6	72	M	AML de novo	Remission induction followed by allotransplantation	Conditioning with fludarabine and treosulfan		+	+	+
7	53	F	AML de novo	Remission induction followed by allotransplantation	Conditioning with busulfan plus cyclophosphamide	+	++	+	
8	57	M	AML de novo	Remission induction followed by allotransplantation	Conditioning with fludarabine and treosulfan		+	+	+
9	51	M	AML de novo	Remission induction followed by autotransplantation	Conditioning with busulfan plus cyclophosphamide			+	
10	43	M	APL relapse	Remission induction followed by autotransplantation	Conditioning with busulfan plus cyclophosphamide		+	+	+
11	34	M	AML de novo	Remission induction followed by autotransplantation	Conditioning with busulfan plus cyclophosphamide	+	+	+	+
**Conventional Induction and/or Consolidation Chemotherapy**									
12	65	F	AML de novo	Induction chemotherapy, regeneration to remission	Induction with daunorubicin plus cytarabine. First consolidation: mitoxantrone, cytarabine. Second consolidation: etoposide, amsacrine and cytarabine	+	++		
13	63	F	AML relapse	Conventional induction, regeneration to remission	Induction with etoposide, amsacrine and cytarabine	+			
14	68	F	AML de novo	Conventional induction leading to remission; later consolidation chemotherapy	Induction: daunorubicin, cytarabine. First consolidation: mitoxantrone, cytarabine.	+	+		
15	18	F	AML de novo	Induction chemotherapy, regeneration to remission	Daunorubicin plus cytarabine	+			
16	64	M	AML relapse	Conventional induction leading to remission; later consolidation chemotherapy	Induction: daunorubicin, cytarabine. First consolidation: mitoxantrone, cytarabine.	+	+		

ALL, acute lymphoblastic leukemia; AML, acute myeloid leukemia; APL, acute promyelocytic leukemia; F, female; M, male; MDS-HR2, myelodysplastic syndrome, high-risk class 2; PLL-T prolymphocyte leukemia T cell type.

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
