# Peer review of "A Pilot Study of Circulating Monocyte Subsets in Patients Treated with Stem Cell Transplantation for High-Risk Hematological Malignancies"

_medicina, 2020, doi:10.3390/medicina56010036_

Round 1
Reviewer 1 Report
In this paper authors show an expected normalization of monocytes after stem cell transplantation.
Line 59: Authors should define “early” reconstitution
Introduction section: please explain your aim and hypothesis, why this research needed? Clinical implications?
Suggest to move lines 124-132 to methods section, start results section by describing the demographics of your patients.
Results section: authors describe 13 patients, but table has 16 patients, I suggest to include only patients that authors studied
Discussion: authors should not only state their finding, but also elaborate its impotence, clinical significance?
Line 315: please avoid using high risk leukemia, a nebulous term, state accepted terminology AML or high-risk MDS
Line 316: “a greater variation in the distribution of monocyte” : greater from what?
the patient cohort is small and heterogeneous; AML, MDS, APL, PLL, also some got autologous most got the allogeneic transplant. it is difficult to draw any reliable conclusion. I suggest authors to focus one disease group and transplant modality.
Reviewer 2 Report
Line 40: Please consider mentioning about the novel therapies in AML as there are subtypes in which intensive chemotherapy and transplant are not the standard.
Line 45: Please mention that the age of transplant depends on the institution and the fraility of the patient.
Line 55: Please be more specific about CD14 role (more than surface receptor).
Line 120: Please be more specific about your statistical analyses.
Table 1: Please include cytogenetic and/or mutations if available.
Table 2: Please change Table 2 into a swimmers plot.
Figure 2: Please remove Figure 2 or offer a more clear representation.
Round 2
Reviewer 1 Report
authors responded my comments